# Milk Protein Adsorption on Metallic Iron Surfaces

**DOI:** 10.3390/nano13121857

**Published:** 2023-06-14

**Authors:** Parinaz Mosaddeghi Amini, Julia Subbotina, Vladimir Lobaskin

**Affiliations:** School of Physics, University College Dublin, Dublin 4, D04 V1W8 Dublin, Ireland; parinaz.mosaddeghiamini@ucdconnect.ie

**Keywords:** nanoparticle, potential of mean force, protein adsorption, protein corona, bio–nano interface, multiscale modeling

## Abstract

Food processing and consumption involves multiple contacts between biological fluids and solid materials of processing devices, of which steel is one of the most common. Due to the complexity of these interactions, it is difficult to identify the main control factors in the formation of undesirable deposits on the device surfaces that may affect safety and efficiency of the processes. Mechanistic understanding of biomolecule–metal interactions involving food proteins could improve management of these pertinent industrial processes and consumer safety in the food industry and beyond. In this work, we perform a multiscale study of the formation of protein corona on iron surfaces and nanoparticles in contact with cow milk proteins. By calculating the binding energies of proteins with the substrate, we quantify the adsorption strength and rank proteins by the adsorption affinity. We use a multiscale method involving all-atom and coarse-grained simulations based on generated ab initio three-dimensional structures of milk proteins for this purpose. Finally, using the adsorption energy results, we predict the composition of protein corona on iron curved and flat surfaces via a competitive adsorption model.

## 1. Introduction

The biological activity and biocompatibility of inorganic materials is controlled by interactions at the bio–nano interface—a nanoscale layer where the material comes in contact with biomolecules. Through biomolecule adsorption, change in protein conformation, and surface chemistry, the bio–nano interface plays a key role in medical applications and food processing. Understanding the underlying mechanism of these interactions may help to control biofilm growing, fouling, or contamination by metabolites arising from bacterial activity on the contact surfaces [1,2,3,4]. Therefore, a molecular-level insight into the main contributions governing the adhesion of biomolecules and the structure of the bio–nano interface would be beneficial.

Among the materials used in industrial and medical applications, one of the most common ones is stainless steel, predominately consisting of metallic iron alloyed with chromium and carbon. These materials are widely used due to low cost, relative biological neutrality, and corrosion resistance at standard low-heat conditions. However, this might be not true at the higher temperatures, e.g., milk fouling and milk contamination is a critical problem in the heating process for the food industry [5]. Daily, about a billion liters of milk is processed. This causes fouling of the surfaces of heat exchangers, which in turn leads to decreased heat transfer efficiency and frequent pressure drop events. Efficient and fast cleaning of factories’ heat transfer equipment is a costly and challenging task. When done often, it can result in reduced production volumes and economical losses [6]. When liquid milk is heated, several processes take place that result in the deposition of minerals and proteins on the surface of containers. Generally, a thin protein layer is formed on a material’s surface right after the liquid milk is brought in contact with a food processing unit [7]. Adsorbed proteins may contribute to the decrease in heating by forming an insulating layer between the heater and material, which can affect milk pasteurization and sterilization and may cause contamination [8]. Physicochemical characteristics of the heat exchanger surface, as well as the conditions and biochemical composition of the milk, determine the outcome of this interaction [9,10].

Besides the food industry, biomolecular interactions on metallic surfaces are essential for implants and other medical devices. When a tissue makes contact with such surfaces, unfavorable reactions such as infections, fibrosis, thrombosis, and inflammation may result [11]. Recent research demonstrates that quickly after implantation, a layer of proteins forms interstitial fluids covering biomaterial implants. Thus, the nature of this protein layer and its properties play a key role in how cells respond. To improve the operation and safety of medical implants, it is imperative to develop materials that cause the necessary tissue reactions. In order to increase implant biocompatibility, the majority of research has concentrated on the engineering of surface features that can change the numbers and types of bound proteins, as well as the conformation, orientation, or binding strength of the adsorbed proteins [11,12,13].

Finally, the interest in bio–nano interactions is also driven by the concerns of the safety of nanoparticles (NPs) for human and animal health. The NP toxicity correlates to chemical aggressiveness of the material and scales with its physicochemical properties, such as surface area, charge, or reactivity [14]. Establishing the relationship between metal properties and the reaction with different biomolecules is therefore key to screening the materials for potential health risks. In practice, the safety assessment is often costly and time-consuming and includes animal studies. In silico modeling may help predict the interactions of nanomaterials with living organisms and provide the required information in a humane and cost-effective way [15,16,17,18,19]. Statistical data-driven methods are used for this purpose, where sufficient data are available [20,21,22]. Recently, physics-based models have also addressed the bio–nano interface. In particular, the mechanisms of the formation of NP protein corona have been studied my multiple labs [23,24,25,26]. It is expected that the composition and configuration of the corona determines the biochemical reactivity and sensitivity of NPs, as well as their cell uptake and systemic transfer [23,27]. Yet, to allow predictive modeling, one needs more information about the interactions at the bio–nano interface and their relation to the material and protein properties.

In this work, we investigate interactions between milk proteins and three face-centered cubic (fcc) slabs of zero-valent iron, constituting a simplified model of stainless steel. To accomplish this, we selected six of the most abundant proteins from the major protein groups found in natural cow milk [28]. Our primary goal is to quantify the binding affinity of these proteins on iron surfaces by evaluating their adsorption energy at their different orientations in respect to solid iron. To calculate these energies and to predict the composition of the insulating protein layer on the metallic surfaces, we invoke a three-stage multiscale computational method, including all-atom [29,30] and coarse-grained (CG) united atom (UA) [21,26,31] and kinetic Monte Carlo (KMC) [22] simulations. This method was previously applied to explain various examples of interfacial phenomena between biomolecules and inorganic materials, including nanotoxicity [32,33]. The remainder of this paper is organized as follows. In the “Materials and Methods” section (Section 2), we provide a detailed explanation of the theoretical model built to study the protein–metal interaction and the rationale behind the model parameterization scheme. In Section 3, we discuss simulation results, analyze individual adsorption affinities predicted for molecules representing the bio part of the interface (amino acids, milk proteins, and carbohydrates), and report their preferred orientations. We also discuss kinetics of competitive adsorption for the six most abundant milk proteins to understand the process of protein deposition on metallic surfaces. Obtained results will be also compared with existing experimental data on protein adsorption. Finally, in Section 4, we summarize the key insights gained from this study.

## 2. Materials and Methods

We aim to calculate the content of milk protein layers on iron surfaces using first principles multiscale simulations. To do so, we use a CG kinetic Monte Carlo (KMC) method [22] to model competitive adsorption of the six most abundant milk proteins. The KMC simulation requires the knowledge of individual binding energies at different orientations (heatmaps) for each selected protein immobilized at each fcc configuration of the metallic surface. Heatmaps for individual proteins can be obtained by UA simulations, as described in a following subsection. Although the UA method was already parameterized for the range of crystalline surfaces for noble metals (Ag, Au, and Cu), oxides (TiO${}_{2}$, SiO${}_{2}$, and Fe${}_{2}$O${}_{3}$), organic NPs (graphene, carbon nanotubes, and carbon black), and semiconductors (CdSe) [34,35], the method is missing the set of short-range potentials essential for calculating milk protein–iron adsorption energies. These potentials in a form of tabulated potentials of mean force (PMF) are calculated from explicit all-atom simulations through a previously introduced scheme [21,29,31].

### 2.1. Protein–Solid Surface Interaction in UA

Generally, the interaction between a protein and a solid surface, such as an inorganic engineered material, includes several contributions, both specific and nonspecific, and depends on the chemical composition, size, shape, surface roughness, surface charge, surface functionalization, and hydrophobicity of the material. All these aspects should be considered for building CG models of the bio–metal interfaces. Various examples of such models (including the UA model) have been previously used to study the competitive adsorption of proteins onto solid surfaces [26,36].

The UA model, previously developed in our lab [21,26,31], includes all major non-covalent contributions to the interactions between the NP and the protein in a simplified way. The solid surface can be represented by a rigid flat (slabs), spherical (NP), or cylindrical (nanotubes) shape. Proteins are considered as rigid-body structures consisting of 20 different amino acids (AA). Each AA is represented by one bead in the UA model, and respective positions of these beads within the protein are fixed. The center of the AA bead is positioned on the AA’s Cα atom. The CG model of the protein has the same three-dimensional structure and key details as the original all-atom model. However, fewer beads will decrease the dimensionality of the model and result in lower calculation times. This model is illustrated in Figure 1 along with the presentations on all-atom and CG UA. Further coarse-graining is applied to reduce the dimension of modeling the kinetics of the competitive adsorption. The ultra-CG hard-sphere (HS) model for selected proteins is parameterized based on the UA heatmaps.

The UA model divides the NP into core and surface segments according to the distances between them and the protein. Then, the interaction potential between each AA and the NP can be represented by a combination of a short-range surface van der Waals (vdW) potential ($U_{s}^{vdW}$), a long-range core vdW potential ($U_{l}^{vdW}$), and an electrostatic potential ($U^{el}$). The interaction potential between the NP and the entire protein ($U_{p-NP}$) is written in a pairwise additive way via interaction potentials for individual AAs with the NP:(1)$$U_{p-NP}=\sum_{i=1}^{N_{AA}}U_{i}\left(d_{i}\left(\theta ,\phi \right)\right)=\sum_{i=1}^{N_{AA}}U_{i}^{el}\left(d_{i}\left(\theta ,\phi \right)\right)+\sum_{i=1}^{N_{AA}}U_{i}^{vdW}\left(d_{i}\left(\theta ,\phi \right)\right)$$
This potential depends on the distance $d_{i}$ between the centers of mass (COMs) of the NP and each AA in the protein. This distance is determined by the protein’s overall orientation relative to the surface of the NP, which is set by two rotational angles (θ,ϕ) with respect to the protein’s initial orientation, as defined in the PDB file. The electrostatic interaction between NP and AA is described by the screened Coulomb potential:(2)$$U_{i}^{el}\left(d_{i}\left(\theta ,\phi \right)\right)=\sum_{j=1}^{N_{e}}l_{B}k_{B}Tq_{i}q_{j}\frac{e^{-\kappa r_{ij}}}{r_{ij}}$$
where $r_{ij}$ is the distance between the residue of charge $q_{i}$ and the point charge $q_{j}$ on the NP surface in terms of the elementary charge $e_{0}$, $k_{B}$ is the Boltzmann constant, *T* is the temperature, $\kappa =\sqrt{4\pi l_{B}I}$ is the Debye length, $l_{B}=\frac{e_{0}^{2}}{4\pi \epsilon \epsilon_{0}k_{B}T}$ is the Bjerrum length, and *I* is the ionic strength $I=\frac{1}{2}\sum_{i}^{N_{ions}}c_{i}z_{i}^{2}$, with $c_{i},z_{i}$ being the ion concentrations and valencies, respectively. The properties of the solvent are reflected in the dielectric constant ϵ. The VdW potential, representing combined dipole–dipole and dispersion interactions between the *i*-th AA and the NP, includes short-range and long-range terms:(3)$$U_{i}^{vdW}\left(d_{i}\left(\theta ,\phi \right)\right)=U_{i,s}^{vdW}\left(d_{i}\left(\theta ,\phi \right)\right)+U_{i,l}^{vdW}\left(d_{i}\left(\theta ,\phi \right)\right)$$

We extracted the short-range surface potentials from all-atom adaptive well-tempered metadynamic (AWR-MetaD) simulations by following the procedure described in the next section. The long-range term arising from the vdW forces acting through the water medium between the core of the NP and the *i*-th AA can be approximated by the Hamaker procedure:(4)$$\begin{array}{c} U_{i,l}^{vdW}\left(R_{NP},R_{AA},d\left(\theta ,\phi \right)\right)=-\frac{A_{123}}{12k_{B}T}\left(\frac{4R_{NP}R_{AA}}{d_{i}{\left(\theta ,\phi \right)}^{2}-{\left(R_{NP}+R_{AA}\right)}^{2}}+\right.\\ \left.\frac{4R_{NP}R_{AA}}{d_{i}{\left(\theta ,\phi \right)}^{2}-{\left(R_{NP}-R_{AA}\right)}^{2}}+2\ln\frac{d_{i}{\left(\theta ,\phi \right)}^{2}-\left(R_{NP}-R_{AA}^{2}\right)}{d_{i}{\left(\theta ,\phi \right)}^{2}-{\left(R_{NP}+R_{AA}\right)}^{2}}\right) \end{array}$$
Here, the coefficient $A_{123}$ corresponds to the interaction of material 1 with material 3 through the medium 2; $R_{AA}$ and $R_{NP}$ are the radii of AA and NP, respectively.

We then calculate the interaction energy for all possible orientations of the protein, described by orientation angles θ and ϕ with respect to the initial orientation (taken from the origin PDB file). The total potential energy is found as a function of distance of the protein center of mass from the NP surface.

Integration of the interaction potential over all possible orientations $\left(\theta_{k},\phi_{l}\right)$ and corresponding distances $0\leq z\leq a(\theta_{k},\phi_{l})$ gives the mean interaction energy $E(\theta_{k},\phi_{l})$. For flat surfaces, the energy is calculated as
(5)$$E\left(\theta_{k},\phi_{l}\right)=-k_{B}T\ln\left[\frac{1}{a(\theta_{k},\phi_{l})}\int_{0}^{a(\theta_{k},\phi_{l})}\exp\left[\frac{-U_{p-NP}\left(z,\theta_{k},\phi_{l}\right)}{k_{B}T}\right]dz\right]$$
For a protein interacting with a spherical NP,
(6)$$E\left(\theta_{k},\phi_{l}\right)=-k_{B}T\ln\left[\frac{3}{{\left(R_{NP}+a\left(\theta_{k},\phi_{l}\right)\right)}^{3}-R_{NP}^{3}}\int_{R_{NP}}^{R_{NP}+a\left(\theta_{k},\phi_{l}\right)}\exp\left[\frac{-U_{p-NP}\left(r,\theta_{k},\phi_{l}\right)}{k_{B}T}\right]r^{2}dr\right]$$

The set of rotational configurations along with the corresponding $E(\theta_{k},\phi_{l})$ is stored in the heatmap. The next step is to calculate the average adsorption energy by using the potential energy as a function of distance for each angle with Boltzmann averaging and weighting factors:(7)$$E_{ads}=\frac{\sum_{k}\sum_{l}P_{kl}E\left(\theta_{k},\phi_{l}\right)}{\sum_{k}\sum_{l}P_{kl}},$$
(8)$$P_{kl}=\sin\left(\theta_{k}\right)\exp\left[-\frac{E(\theta_{k},\phi_{l})}{k_{B}T}\right],$$
where $P_{kl}$ is the Boltzmann weighting factor. Below, we show how the maximum binding energy affecting the protein adsorption the most can be defined from the adsorption heat maps.

### 2.2. All-Atom Model for Recovering Short-Range Potentials

The interaction of atoms with a solid surface can be best performed using enhanced sampling methods [37,38,39,40]. These techniques include estimating the free energy during simulation and feeding that information back into the dynamics as a statistical bias. The common goal is to spend as little time as possible sampling already sampled free energy zones [41]. Adaptive biasing force methods, which employ an approximation of the mean force to bias the dynamics, are one of the strategies that can speed up the simulation [42,43]. In this work, we use the adaptive well-tempered metadynamics method that has previously been described for measuring the adsorption of the biomolecules for TiO${}_{2}$ [29] and Ag [30]. The approach keeps the adsorbate away from previously visited regions of the collective variable space by introducing a time-dependent bias to the system’s potential energy. GROMACS-2018.6 software was used to carry out the simulations in this section of the work [44,45]. A CHARMM-GUI/Nanomaterial Modeler was used to create three fcc surfaces of iron: (100), (110), and (111) [46].

The AA side chain analogues (SCAs are the compounds that result from cutting off the side chains of AAs at the protein backbone and substituting a hydrogen for the carbon) were placed in the center top of the slab. In this work, we calculated the energy for 22 SCAs, including two forms of Histidine, HID, and HIE, depending on the location of the protonation on the nitrogen atom and two forms of glutamic acid, GLU and GAN, negatively charged and neutral, respectively (see the Appendix A for more details). The system was solvated using the original form of TIP3 water model [47]. It was then neutralized by NaCl regarding the charge of the whole system. In the MD calculations, the energy of the system was minimized by Verlet particle-based cut-off scheme using charge groups and steepest gradient method for 1000 steps. Figure 2 shows the simulation box, the Fe-100 slab, and the SCA after the minimization, the water molecules were removed for better visualization. The system was equilibrated under constant pressure 1.0 bar and temperature 300 K conditions (NPT ensemble) using Berendsen weak coupling [48] for 1.0 ns. Then, the system was pre-equilibrated for 10 ns in the NVT ensemble. The Nose–Hoover thermostat’s relaxation time constant for the NVT ensemble was 5 ps. The cut-off distance was set to 1.0 nm for the VdW short-range interactions. In the metadynamic-biased simulation, the surface separation distance (SSD) was used as the reaction coordinate. This SSD measures the minimum distance of the COMs of the SCA $R_{mol}$ and the surface atoms $r_{i}$ along the *z* axis,
(9)$$SSD=\min|R_{mol}-r_{i}|,$$

In the adaptive well-tempered metadynamics, the adsorption energy is measured as a function of SSD. These simulations were conducted in the NVT ensemble with the PLUMED (PLUMED2-2.5.1.conda.5) software plugin in GROMACS [44,45,49]. The simulation time was between 500–600 ns for each SCA to obtain the set of short-range one-dimensional PMFs in tabulated form. The bias factor and the time factor were set to 10 and 500 ps, respectively. Every 0.5 ps, the Gaussian was added, the initial height was 1.75 kJ/mol, and the temperature was kept at 300 K.

### 2.3. Preparation of Starting Coordinates for Biomolecules and Surfaces

We simulated three fcc surfaces of iron slabs (100, 110, and 111) using GROMACS with the CHARMM-GUI/Nanomaterial Modeler tool and force fields [46,50,51]. The slab thickness for all surfaces (100, 110, and 110) was between 3.53 and 3.71 nm, elongated in the *z* direction in vacuum, and was in periodic boundary conditions along the *x* and *y* coordinates. The final simulation box sizes were 3.572×3.572×9.127 nm${}^{3}$ for Fe-100, 3.548×3.561×9.914 nm${}^{3}$ for Fe-110, and 3.541×3.943×9.258 nm${}^{3}$ for Fe-111. To calculate the PMFs at the atomistic level, 22 sets of SCAs were chosen, which are sufficient to assess adsorption affinities for different kinds of proteins in the CG UA model.

In order to investigate the interaction of the entire protein with the NP, the UA model reads the atomistic 3D structure of the protein. These structures might not be available from experiments such as X-ray crystallography, NMR spectroscopy, or cryo-electron microscopy. Thus, we predicted 3D structures of the proteins using the sequence data. Here, we used I-TASSER (Iterative Threading ASSEmbly Refinement) 5.1 software [52] to generate 820 milk protein structures [53,54]. It should be noted that the majority of these proteins do not exhibit a well-defined globular structure. Table 1 shows the list of 6 selected representative milk proteins, UniProt ID, molecular weight, charge, and number of AAs in each molecule. The charge information was obtained from PROPKA calculation [55,56] at pH 7.0. All the proteins were subsequently equilibrated in water for 50 ns under NVT and NPT ensembles (see Appendix A).

To model competitive protein adsorption from milk, we imitated the natural protein concentrations. The majority of the milk ingredients are water (86–88%), fat (3–6%), protein (3–4%), lactose (5%), and minerals (0.7%) [28]. Milk proteins have many physicochemical properties that allow them to be used in a wide range of applications, from nutritional to functional and biological functions [57]. Caseins and soluble (whey) proteins are classified into two broad categories of milk proteins. Caseins account for about 78 % of bovine milk proteins in total and whey proteins for 17% [58]. Caseins can be classified into four genetic groups based on the similarity of their primary AA sequences [28]. These four groups are αs${}_{1}$-caseins, αs${}_{2}$-caseins, β-caseins, and κ-caseins and correspondingly make up 38%, 10%, 36%, and 13% of the total casein [59]. In decreasing quantities, the whey proteins consist of β-lactoglobulins, α lactalbumins, immunoglobulin, and serum albumin, which represent 60%, 20%, and 10% of total whey proteins [60,61]. In the competitive adsorption simulations, we set the representative protein concentrations in solution according to their relative mass concentrations in milk. We estimated this concentration based on the the percentage of each protein and considering the fact that cow milk has 30–39 g/L of protein in total. The molar mass of each protein was taken from AlphaFold database [62].

### 2.4. Competitive Adsorption Model

With extracted individual protein UA adsorption heatmaps, we can predict the composition of the adsorbed protein layer (NP protein corona) on the metallic surface. This layer can be subdivided into hard and soft corona [63]. The hard corona consists of proteins that are bound directly to the surface of the material with strong affinity, and soft corona includes proteins that are weakly bound on top of the hard corona via protein–protein interaction [22,63]. We further study the hard corona layer. The composition of the hard corona evolves with time and depends on the morphology and composition of the adsorbent material. In this dynamical process, which can last several hours in real systems, proteins diffuse from the bulk of the solution and adsorb on solid surfaces before occasionally desorbing and being replaced by other proteins. Since the milk sample contains hundreds of proteins, all-atom and even UA simulations of the simultaneous adsorption of all milk proteins would be very time-consuming [64]. Here, we use an ultra-CG method to describe the actual adsorption–desorption dynamics, representing each protein as a single ultra-CG bead. Since the adsorption rate of incoming proteins is dependent on previously adsorbed molecules, the whole surface area of the metal will not be accessible for full coverage by proteins [65]. The HS model, which was initially developed for predicting adsorption onto NPs, considers proteins as rigid spheres with established positions, while the solid is modeled as a spherical NP. In this model, the protein is in a reversible physical contact with the NP’s surface and occupies space that becomes unavailable to other proteins. For each protein, we set a molar concentration in solution, as well as rate constants for adsorption and desorption. Due to the slow rates of protein adsorption and desorption, we use reaction–diffusion equations and the KMC method to simulate the process [22]. The corona formed by milk proteins on a large iron NP predicted using KMC calculations is discussed in Section 3.4.

## 3. Results and Discussion

### 3.1. Short-Range Potentials for SCAs on Iron Surfaces

The short-range potentials (in $k_{B}T$) for 22 AA SCAs on Fe (100, 110, and 111) obtained by atomistic MD with metadynamics are shown in Figure 3. Water density profiles from MD simulations for the slab–water system indicate that two water layers with elevated density are formed near the surface at around 0.2 nm and 0.6 nm for all three fcc surfaces. We can identify three different regions: one close to the wall with strong repulsion, an intermediate region between the two layers of water, and the bulk region when the PMF curve comes to a plateau. Traditionally, AAs are classified into four groups as hydrophobic, charged, polar, and aromatic. According to our data, these groups demonstrate different binding affinities to the zero-valent iron surfaces. Hydrophobic AAs (such as ALA, VAL, and LEU) bind weakly, while aromatic and charged AAs (such as PRO, TYR, ARG, and HIS) are more strongly attached. Figure 4 provides a comparison of the binding energies for different AAs.

Analysis of the PMFs suggests that the deepest minimum for SCA adsorption is located close to the first water adlayer, which can facilitate the adsorption of hydrophilic and charged AAs. Similar behavior was reported for adsorption of AAs onto noble metal surfaces [30,66,67,68,69,70], with the only difference being with sulfur-containing AAs. For example, for the most-studied AuNPs, the binding affinity ranking was as follows: S-containing > aromatic > amines > aliphatic > amides > hydroxylic > carboxylic. The energies of adsorption for individual SCAs ranged from $-0.35k_{B}T$ to $-39.69k_{B}T$ (see Appendix A for more details).This predicted range of adsorption energies for the small molecules in our study is in line with the adsorption free energy of furfural onto zero-valent iron NPs reported in [71]. For ALA SCA (whose model is chemically equivalent to methane) onto Fe-110, we found adsorption energy close to $0k_{B}T$, corresponding to the lack of binding. A similar adsorption pattern was reported in [72], where no stable binding complexes were predicted for CH${}_{4}$ by DFT calculations.

### 3.2. Protein Adsorption Energies, Preferred Orientations, and Heatmaps

To model milk protein adsorption at the metallic surfaces, we selected the six most abundant proteins, listed in Table 1. These proteins were subjected to UA model simulations to obtain their adsorption binding affinities and preferred orientations for the protein immobilized on the iron surfaces. For these calculations, we used $R_{NP}=80$ nm, with zeta potential −5 mV at pH 7.0. We report a single value for each protein after simple averaging the energy over all possible orientations. The obtained adsorption energy rankings based on the lowest energy values of the adsorption heatmaps are listed in Table 2. AS1C and AS2C were predicted to be the most strongly bound proteins, while BLAC and ALAC are the weakest bound proteins on all Fe (100, 110, and 111) surfaces. The preferred orientation of the proteins would not change that much on different fcc facets of iron.

The preferred orientations of all 820 milk proteins based on the lowest energy from heatmaps are reported in the Appendix A. We performed calculations with 5 Fe NPs of radii 5 nm, 20 nm, 50 nm, 80 nm, and 100 nm to study the size dependence of adsorption energies. The data are presented in the Appendix A. Figure 5 shows the output of the UA model for the six selected milk proteins on iron NPs. The heatmap contains the adsorption energies for all values of θ and ϕ. The blue spots (with lower energy) correspond to the more favorable orientations of immobilized proteins. A 3D presentation of the protein on the NP surface is shown on top of each heatmap corresponding to the same protein. The closest AAs to the surface of the NP are marked in each case. Analysis of the configurations shows that LYS, GLU, PRO, and ASN are the most likely to be in contact with the metal surface. This observation is in agreement with the results from ref. [73] that are discussed in Section 3.3.

The protein adsorption energies predicted by the UA model (Table 2) are high and correspond to the irreversible binding of these molecules. This is a result of a high activity of zero-valent Fe. For example, for the BSA protein, the average adsorption energy for all fcc facets was $-23.67k_{B}T$ for zero-valent Ag [30] and $-21.68k_{B}T$ for zero-valent Au [32,74] vs. $-75.23k_{B}T$ for zero-valent Fe. Please refer to Appendix A for the comparison of heatmaps depicting the interaction of each milk protein on Fe-100, Fe-110 and Fe-111 surfaces.

### 3.3. Validation of UA Model Parameters

To validate our model, we further considered interaction between zero-valent FeNPs and human serum albumin (HSA) through both experimental and docking simulation methods. Analysis using fluorescence spectroscopy showed that the FeNPs formed a complex with HSA through hydrogen bonds and van der Waals interactions. Furthermore, circular dichroism spectroscopy showed that the secondary structure of HSA was not affected by the FeNP. An MD study indicated that the FeNPs interacted with polar residues on the surface of the HSA molecule. The docking study found that ASP, ARG, SER, LYS, and GLU residues are the most likely to be on the FeNP surface. The reported adsorption free energy *G* was −204.80 kJ/mol, which is comparable to our observations. Taking the same initial protein PDB structure and temperature and size for Fe-110 in the UA model, we measured that the minimum free energy was equal to −317.84 kJ/mol at $\theta =50^{\circ }$ and $\phi =340^{\circ }$. The nearest interacting AAs in our case are LYS-564, GLU-565, LYS-573, GLU-505, ASP-562, GLU-82, ARG-81, LEU-80, ASP-56, and ASP-129 that are shown in Figure 6.

In reality, the metallic iron surfaces are quickly oxidized and hydroxylized, which changes their adsorption affinity to water and reduces the binding strength for AAs and proteins. Another assumption that affects the range of the observed energies is that the orientation distribution for adsorbed proteins is equilibrium. This assumption is used implicitly in the ensemble averaging procedure represented by Equation (Equation 7). This average energy is dominated by a single preferred orientation. If, however, we assume that proteins bind in random orientations, then the simple average over all angles would reflect the actual binding strength. Yet, in a previous study [30], we found that the Boltzmann average correlates better with the adsorption affinity ranking for proteins, so we used these figures below to analyze binding affinities of individual proteins and to model the competitive adsorption. Appendix A discusses another experimental approach to validate the binding properties of Hen egg-white lysozyme (HEWL) on zero-valent FeNP (see Appendix A).

### 3.4. Competitive Adsorption and Milk Protein Layer

We next determined the composition of the milk protein layer at the iron surfaces. As discussed above, in this method, we assumed that the surface is represented by a spherical NP with the protein layer adsorbed on the whole surface of the NP (protein corona), and the adsorption energy is determined by the UA method. The adsorption free energy controls the behavior between the NP and the protein by using the HS KMC model. The higher the adsorption energy, the more probable it is for the protein to be bound on the surface for long periods of time. Calculating the actual adsorption kinetics for different proteins is more challenging. After giving the system enough time, the proteins will compete and form a stable protein corona around the NP. The concentrations of milk proteins to determine the corona were chosen from the literature [28,59]. The adsorption–desorption process stabilizes after a certain time, which depends on the NP radius.

Figure 7 shows a snapshot of the six milk proteins adsorbing on the iron surface from a solution imitating natural milk. Our estimates suggest that the layer of the milk proteins should come to equilibrium within about 40 min, due to the high individual adsorption affinities for selected proteins. The process stops when the surface reaches full loading capacity with no more space left for further protein adsorption and the protein replacements cease. We can see from the Figure 7 and Table 3 that AS1C (orange-colored beads) is adsorbed in the largest numbers on the surface, and BSA (brown-colored beads) has the least abundance in the corona. The left panel of Figure 7 demonstrates the nontrivial kinetics of the corona formation: the concentrations of bound BC, BLAC, and ALAC decrease with time after a very quick buildup, while the concentration of AS2C grows much more slowly, only to exceed the amount of the former after about 0.3 h, evidently replacing the originally bound proteins. The concentration of AS1C remains high after the initial buildup, and the concentration of BSA stays low at all times.

We repeated the KMC simulations several times for each NP size and averaged the observables over at least three randomly selected runs. Results of these simulations are listed in Table 3. There, we present the average abundance per unit area and relative mass abundance for each milk protein on each surface of fcc Fe. The results in the table show that AS1C is the most abundant, both in terms of number of molecules and mass, on all the iron surfaces, while BSA has the smallest number and mass abundance. BLAC and BC are almost equally present in the corona and are the second and third most abundant proteins. ALAC and AS2C, respectively, are the forth and fifth most abundant proteins. This is explained by the relatively low molar fraction of BSA in milk as compared with caseins and lactalbumins.

The corona results obtained on NPs of $R_{NP}=80$ nm essentially represent flat iron surfaces. In previous works [31], we demonstrated that for the majority of proteins, the adsorption energies and preferred orientations for most proteins do not change after an NP radius of about 30 nm, as the surface curvature becomes too large compared with the protein size. Therefore, we expect that the predicted abundances and amounts of deposited proteins should be valid for industrial applications involving steel or iron devices. In the future, we are planning to extend the analysis to competitive adsorption involving lactose and milk fat at different temperatures. Further extension of this work is possible with an inclusion of protein unfolding and denaturation at high temperatures and different pH values. This, however, requires much more complex simulations with flexible proteins and 3D structures adjusted to the specific conditions and may be currently unfeasible in the most detailed form.

## 4. Conclusions

In this work, we presented the results of modeling the interaction between iron surfaces and the most abundant milk proteins in a multiscale scheme based on a combination of the CG UA model with the KMC model for predicting the protein corona composition in the deposited milk layer on iron surfaces. We considered a simplified model of milk formed from the solution of the six most abundant proteins occurring in natural cow milk. Our study ranked the proteins by adsorption strength as follows: α s${}_{1}$-caseins, αs${}_{2}$-caseins, β-casein, bovine serum albumin, α-lactalbumin, and β-lactoglobulins, respectively, from the strongest to the weakest binding. We found that the amount of the bound protein depends on its concentration in solution, so the KMC simulation of the adsorption kinetics ranked the proteins differently (in terms of mass fraction in the milk model solution): αs${}_{1}$-caseins, β-lactoglobulins, β-casein, α-lactalbumin, αs${}_{2}$-caseins and, finally, bovine serum albumin. In the future, we are planning to extend the multiscale model of milk adsorption by adding lactose and milk fat molecules.

Our multiscale model of protein corona formation on solid NPs and surfaces can be generalized to a large variety of systems. It essentially relies only on the existence of an atomistic force field for the solids. It can be used for a variety of applications, such as controlling fouling, biofilm growth in food processing and packaging, and medical devices. We also plan to extend the boundaries of the multiscale model to a wider range of temperatures and pH values employed in industrial settings.

## Figures and Tables

**Figure 1 nanomaterials-13-01857-f001:**
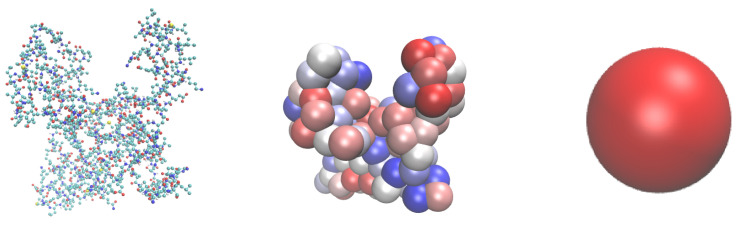
All-atom (**left**) vs CG UA (**middle**) vs single-bead KMC (**right**) structure of bovine β-casein protein (using I-TASSER-predicted structure).

**Figure 2 nanomaterials-13-01857-f002:**
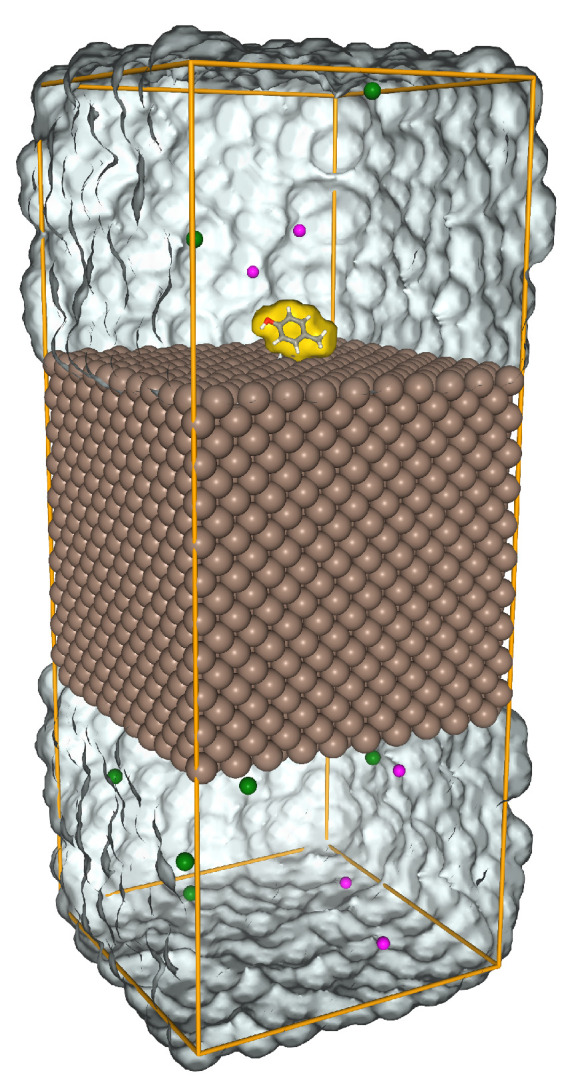
System used to simulate the adsorption of SCAs on Fe slabs.

**Figure 3 nanomaterials-13-01857-f003:**
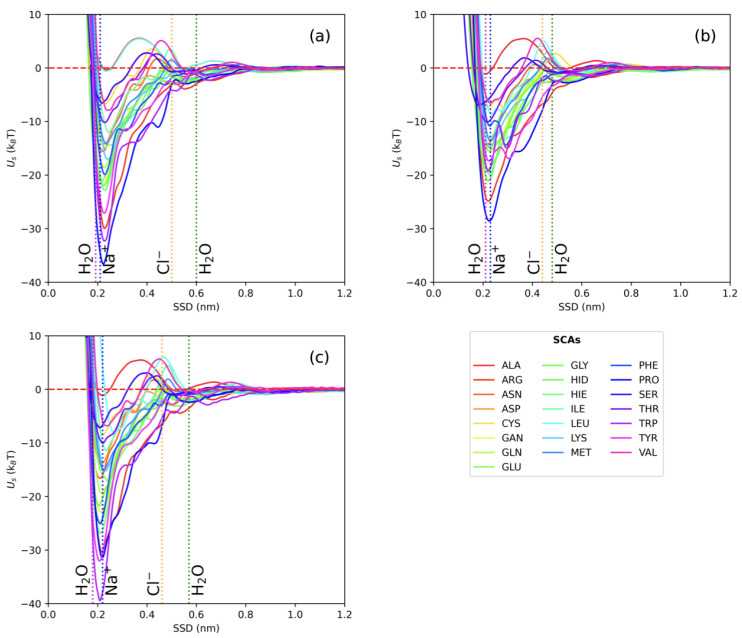
Adsorption free energy profiles of AA SCA on three Fe fcc slabs as a function of SSD, calculated using AWT-MetaD. The vertical lines show the position of water and ion layers: (**a**) Fe-100 (**b**) Fe-110, and (**c**) Fe-111.

**Figure 4 nanomaterials-13-01857-f004:**
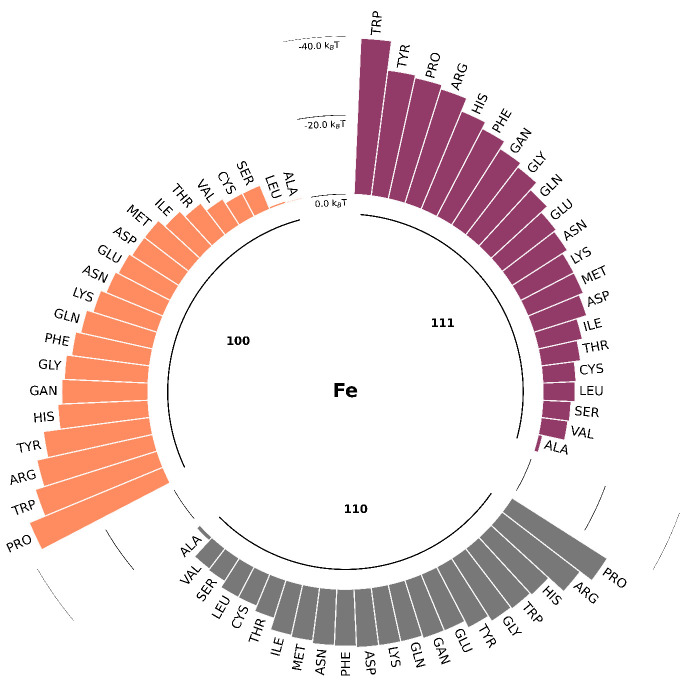
Energy of adsorption (in $k_{B}T$ at the minimum of the PMF) for each SCA on three Fe fcc slabs (100, 110, and 111). Hydrophobic and polar AA weakly bind to the surfaces, while the aromatic and charged AAs are bind the most strongly. Fe-111 shows stronger binding in comparison with Fe-100 and Fe-110.

**Figure 5 nanomaterials-13-01857-f005:**
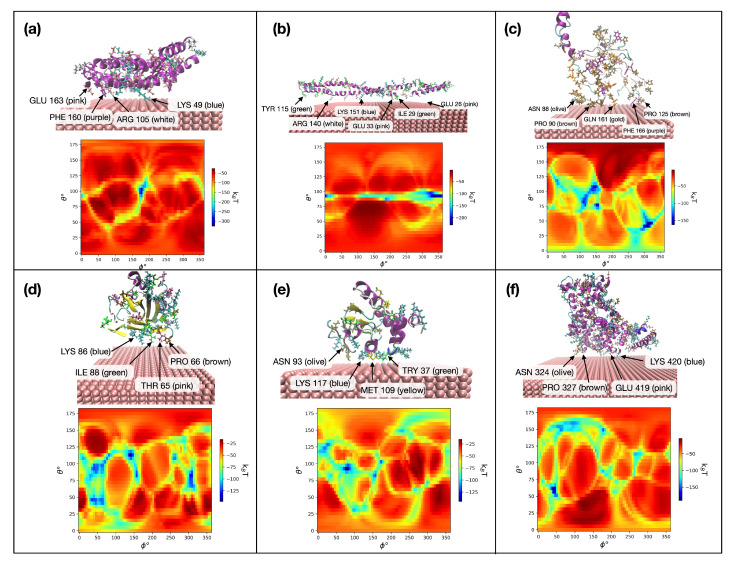
Heatmap and corresponding 3D representations of the interaction of the (**a**) αs${}_{1}$-casein, (**b**) αs${}_{2}$-casein, (**c**) β-casein, (**d**) β-lactoglobulins, (**e**) α-lactalbumin, and (**f**) bovine serum albumin with Fe-110 on the preferred orientation; the figure is showing the closest AAs to the surface of the material.

**Figure 6 nanomaterials-13-01857-f006:**
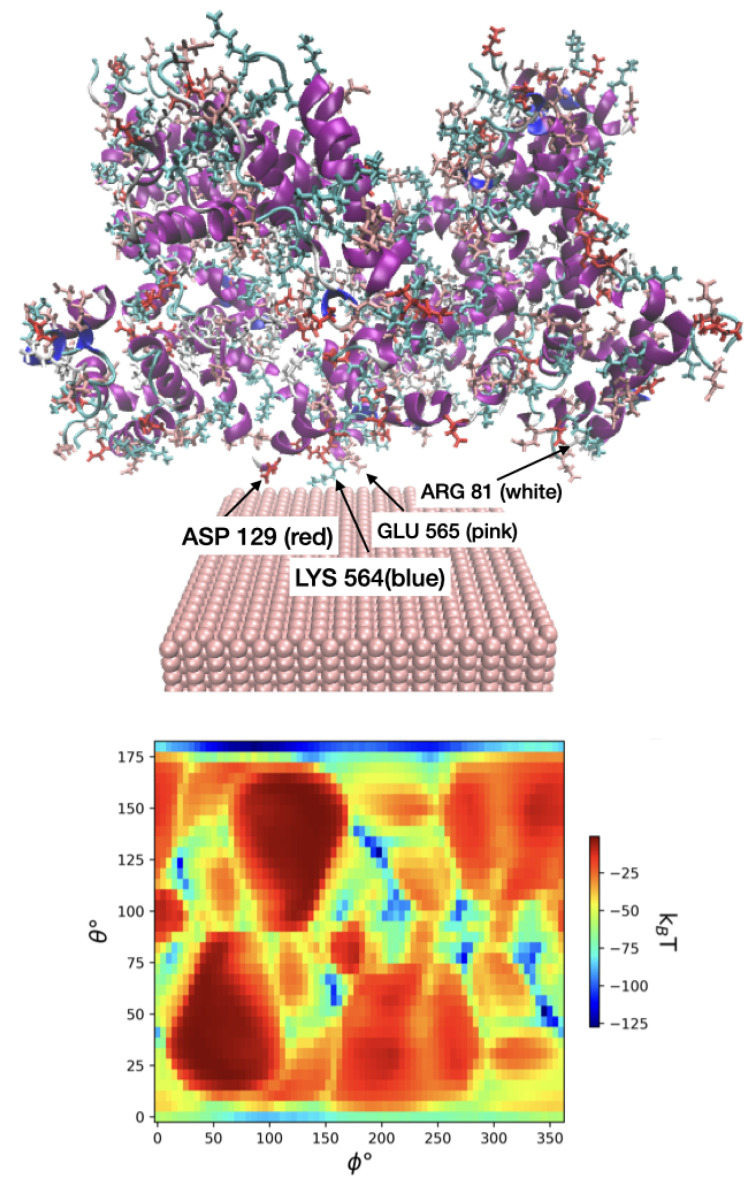
The preferred adsorbed state of HSA on Fe-110 predicted using UA. The nearest AAs of the protein to the slab (LYS, GLU, ASP, and ARG) are labeled.

**Figure 7 nanomaterials-13-01857-f007:**
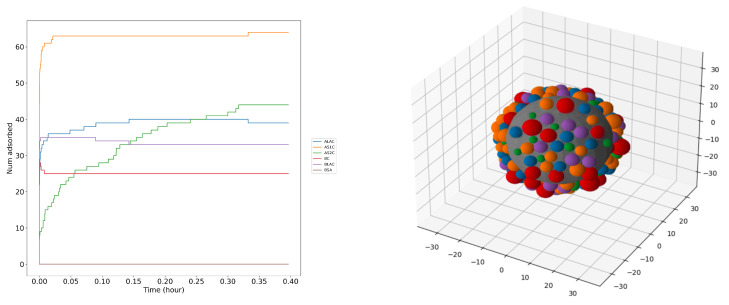
Protein adsorption kinetics on an FeNP plot that shows the adsorption–desorption of each protein over time (**left**). A snapshot of an NP (gray) of radius size 20 nm with protein corona of milk proteins (colored) adsorbing on the surface using the KMC model (**right**).

**Table 1 nanomaterials-13-01857-t001:** Characteristics of the selected milk proteins.

Abbreviation	UniProt ID	Protein Name	$\mathrm{MW}{\;}^{a}$, Da	Charge, *e*	$\mathrm{Res}{\;}^{b}$	$C{\;}^{c}\left[10^{-4}\right]$, mol/L
AS1C	P02662	αs${}_{1}$-casein	24,528.00	−8.5	214	4
AS2C	P02663	αs${}_{2}$-casein	26,018.69	4.5	222	1
BC	P02666	β-casein	25,107.33	−4.5	224	4
ALAC	P00711	α-lactalbumin	16,246.61	−5	142	0.9
BLAC	P02754	β-lactoglobulin	19,883.25	−6	178	2
BSA	P02769	Bovine Serum Albumin	69,293.41	−4.5	607	0.1

(a) Molecular weight, (b) number of residues, and (c) concentrations (mol/L) of the protein in milk that were used in KMC calculations; Section 3.4.

**Table 2 nanomaterials-13-01857-t002:** Comparison of milk proteins’ binding affinities, orientations on Fe-100, Fe-110, and Fe-111, ordered by the binding strength on each surface.

Individual protein adsorption parameters onto Fe-100				
Protein,	$E_{ad}$,$k_{B}T$	θ,${}^{\circ }$	ϕ,${}^{\circ }$	$r_{min}$, nm
AS1C	−343.07	175	100	0.14
AS2C	−238.64	335	90	0.08
BC	−211.96	315	45	0.04
BSA	−202.99	40	60	0.08
ALAC	−159.12	80	90	0.18
BLAC	−142.98	140	110	0.19
Individual protein adsorption parameters onto Fe-110				
Protein,	$E_{ad}$,$k_{B}T$	θ,${}^{\circ }$	ϕ,${}^{\circ }$	$r_{min}$, nm
AS1C	−325.92	175	100	0.14
AS2C	−229.90	335	90	0.07
BSA	−189.87	40	60	0.06
BC	−162.43	310	45	0.04
BLAC	−147.43	140	110	0.18
ALAC	−144.41	75	90	0.16
Individual protein adsorption parameters onto Fe-111				
Protein,	$E_{ad}$,$k_{B}T$	θ,${}^{\circ }$	ϕ,${}^{\circ }$	$r_{min}$, nm
AS1C	−330.36	175	100	0.14
AS2C	−230.99	330	90	0.10
BC	−186.44	310	40	0.02
BSA	−178.25	40	60	0.06
ALAC	−154.60	75	90	0.15
BLAC	−151.56	140	110	0.18

Descriptions of all 820 milk proteins interaction with FeNP surfaces based on the minimum energy of adsorption, their preferred orientation, and minimum distance from surface are available in Appendix A.

**Table 3 nanomaterials-13-01857-t003:** Mean amounts of proteins adsorbed on Fe surfaces per unit area: number concentration (per nm${}^{2}$) and mass abundance obtained from KMC simulations with NPs of radius 80 nm.

	Fe-100	Fe-100	Fe-110	Fe-110	Fe-111	Fe-111
**Protein**	$N_{\mathrm{ads}}[10^{-3}$, **nm**${}^{-2}$]	$M_{\mathrm{ab}}$,%	$N_{\mathrm{ads}}[10^{-3}$, **nm**${}^{-2}$]	$M_{\mathrm{ab}}$, %	$N_{\mathrm{ads}}[10^{-3}$, **nm**${}^{-2}$]	$M_{\mathrm{ab}}$, %
AS1C	9.8	41.22	9.0	38.77	9.6	40.33
BLAC	5.6	19.18	5.5	19.37	5.4	18.50
BC	4.1	17.40	4.2	18.80	4.3	18.29
ALAC	5.5	15.43	5.6	16.29	5.7	15.94
AS2C	1.4	6.19	1.3	6.15	1.4	6.19
BSA	0.05	0.59	0.05	0.60	0.06	0.73

## Data Availability

We have made available all the relevant data for the parameters related to the PMFs of zero-valent FeNP surfaces, I-TASSER PDB structures for 820 milk proteins, and equilibrated PDB structures for the six most abundant milk proteins with MD calculation. The data are publicly accessible at the following links: https://doi.org/10.5281/zenodo.7740918 (accessed on 27 March 2023) (short-range surface potentials (PMFs)), https://doi.org/10.5281/zenodo.6066666 (accessed on 13 February 2022) (i-TASSER structures of the milk proteins) along with https://doi.org/10.5281/zenodo.7798500 (accessed on 4 April 2023) (six MD-equilibrated PDB coordinates). The recent distribution of United Atom and KMC codes can be found here: https://bitbucket.org/softmattergroup/unitedatom.git (accessed on 23 February 2023).

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
