# Peer review of "Milk Protein Adsorption on Metallic Iron Surfaces"

_nanomaterials, 2023, doi:10.3390/nano13121857_

Round 1

Reviewer 1 Report

The manuscript titled "Multiscale modeling of milk protein adsorption on metallic iron surfaces" by Parinaz Mosaddegi Amini et al. presents the findings of a study that employed a multiscale approach, integrating all-atom and coarse-grained simulations, to generate 3D protein structures. By examining adsorption energies, the authors make predictions about the composition of the protein corona on iron nanoparticles and flat surfaces using a competitive adsorption model. The paper reports interesting and well discussed results. However, there are some errors in the manuscript which need to be rectified before the acceptance of the paper. Some suggestions are provided below to improve the manuscript quality.

 In the Results section of the manuscript, although there are explanations for Figures 4 and 6, they are not marked with the corresponding sections of text. Please provide the appropriate annotations for these figures.

 What is the pH value set in the simulation of protein-nanoparticle interactions?

 Table 3: Revise “BS” to “BC”.

Author Response

We thank the reviewer for supporting our paper. We have implemented the suggestions and highlighted them in the revised manuscript.

Reviewer 2 Report

Just accept. The paper is very sound technically and is relevant practically. Possibly such a review (extremely rare from my side) results from a contrast with everything else (of poor quality) that I read this morning, but I do not see any substantial issues.

english is acceptable

Author Response

We thank the reviewer for the support.